# Bias-correcting carbon fluxes derived from land-surface satellite data for retrospective and near real-time assimilation systems

Brad Weir[1,2], Lesley E Ott[2], George J Collatz[2], Stephan R Kawa[2], Benjamin Poulter[2], Abhishek Chatterjee[1,2], Tomohiro Oda[1,2], and Steven Pawson[2]

[1]Universities Space Research Association, Columbia, MD, USA
[2]NASA Goddard Space Flight Center, Greenbelt, MD, USA

**Correspondence:** Brad Weir (brad.weir@nasa.gov)

**Abstract.** The ability to monitor and understand natural and anthropogenic variability in atmospheric carbon dioxide ($CO_2$) is a growing need of many stakeholders across the world. Systems that assimilate satellite observations, given their short latency and dense spatial coverage, into high-resolution global models are valuable, if not essential, tools for addressing this need. A notable drawback of modern assimilation systems is the long latency of many vital input datasets, e.g., inventories,

in situ measurements, and reprocessed remote-sensing data can trail the current date by months to years. This paper describes techniques for bias-correcting surface fluxes derived from satellite observations of the Earth's surface to be consistent with constraints from inventories and in situ $CO_2$ datasets. The techniques are applicable in both short-term forecasts and retrospective simulations, thus taking advantage of the coverage and short latency of satellite data while reproducing the major features of long-term inventory and in situ records. Our approach begins with a standard collection of diagnostic fluxes which incorporate

a variety of remote-sensing driver data, viz. vegetation indices, fire radiative power, and nighttime lights. We then apply an empirical sink so that global budgets of the diagnostic fluxes match given atmospheric and oceanic growth rates for each year. This step removes coherent, systematic flux errors that produce biases in $CO_2$ which mask the signals an assimilation system hopes to capture. Depending on the simulation mode, the empirical sink uses different choices of atmospheric growth rates: estimates based on observations in retrospective mode and projections based on seasonal forecasts of sea surface temperature in

forecasting mode. The retrospective fluxes, when used in simulations with NASA's Goddard Earth Observing System (GEOS), reproduce marine boundary layer measurements with comparable skill to those using fluxes from a modern inversion system. The forecasted fluxes show promising accuracy in their application to the analysis of changes in the carbon cycle as they occur.

## 1 Introduction

As the number and kind of space-based carbon dioxide ($CO_2$) measurements continue to grow, so too do the capabilities of modeling and data assimilation systems which support carbon monitoring. An eventual goal of these systems is the verification of international climate agreements (Ciais et al., 2015; Peters et al., 2017; Pinty et al., 2017; 2019). However, verification would

require the additional ability to distinguish the signal in atmospheric $CO_2$ due to changes in anthropogenic emissions from that due to inter-annual variability in biospheric fluxes, transport, and other natural processes. While this functionality matures, there are many other applications that do not require attribution but do require near real-time (NRT) latencies (i.e., less than a few days), horizontal resolutions finer than several hundred kilometers, and extensive spatial coverage. These include short-term forecasts for field campaigns, identification of times and places of interest for satellite instruments with controllable pointing, (e.g., the Orbiting Carbon Observatory 3), boundary conditions for regional models, and the production of a priori profiles and/or evaluation datasets for retrieval algorithms. Furthermore, high-resolution, global atmospheric $CO_2$ monitoring systems with forecasting and assimilation capabilities enable the study of carbon cycle phenomena as they occur, e.g., the impact of the recent Coronavirus pandemic (Weir et al., 2020), and complement other existing and forthcoming remote-sensing observations of soil moisture, evapotranspiration, and terrestrial biomass (Stavros et al., 2017).

The modeling and assimilation system under development at NASA Goddard Space Flight Center (GSFC), coordinated by the Global Modeling and Assimilation Office (GMAO), incorporates an extensive array of satellite observations and Earth system models to estimate carbon fluxes, atmospheric mixing ratios, and their uncertainties (Ott et al., 2015). The "baseline" configuration of this system features a collection of diagnostic surface fluxes derived from remotely-sensed properties, e.g., terrestrial biospheric exchange from vegetation indices (Randerson et al., 1996), NRT biomass burning from fire radiative power (Darmenov and da Silva, 2015), and space-time disaggregations of fossil fuel inventories from nighttime lights (Oda et al., 2018). High-resolution estimates of atmospheric $CO_2$ then follow from transport simulations with the Goddard Earth Observing System (GEOS) general circulation model, which can reproduce the meteorology of an atmospheric analysis, e.g., the Modern-Era Retrospective analysis for Research and Applications, Version 2 (MERRA-2; Gelaro et al., 2017) with high fidelity (Orbe et al., 2017).

This system is currently being extended to assimilate a collection of $CO_2$ datasets (Tangborn et al., 2013; Eldering et al., 2017) including retrievals of column averages from the Greenhouse Gases Observing Satellite (GOSAT; Kuze et al., 2009), Orbiting Carbon Observatory 2 (OCO-2; Crisp et al., 2004), and the ground-based Total Carbon Column Observing Network (TCCON; Wunch et al., 2011) and in situ data from the multi-agency collection of data provided in NOAA's Observation Package (ObsPack; Masarie et al., 2014). The assimilation step has the benefit of synthesizing heterogeneous measurement types, "gap filling" data when and where they are unavailable, and drawing model simulations closer to observed values. In particular, assimilation systems have the potential to correct for errors in the surface flux inputs associated with uncertainties in how to 1) reconcile bottom-up global budgets derived from inventories and biospheric models with top-down budgets derived from atmospheric measurements (Le Quéré et al., 2018), 2) partition the surface sink between the Tropics and Extratropics (Schimel et al., 2015), and 3) reproduce responses to inter-annual variability in meteorology, for example the impact of El Niño on terrestrial (Liu et al., 2017) and oceanic (Chatterjee et al., 2017) fluxes and drought on semi-arid ecosystems (Poulter et al., 2014). A comprehensive description of current and planned space-based observations of $CO_2$, along with the scientific questions they hope to address, is available in the report of Crisp et al. (2018).

As the first step in the development of the NASA GMAO $CO_2$ assimilation system, we bias-correct the a priori, baseline diagnostic fluxes to match global budget constraints from inventories and in situ measurements. This step reduces the errors of

model simulations before assimilating satellite data, thus increasing the ability of the system to capture other signals of interest in the observations (Dee, 2005). For example, in any given year, estimates of global net biospheric exchange (NBE) from the TRENDY ensemble of terrestrial biosphere models (Sitch et al., 2015) have a range of roughly 4 petagrams of carbon (Pg C). Transported through the atmosphere, this produces a 2 ppm spread in global $CO_2$, which is twice the size of the atmospheric growth rate perturbation of the 2015–2016 El Niño. An under/over-estimation of NBE can lead to an under/over-prediction of the seasonal cycle amplitude of simulated $CO_2$ (Yang et al., 2007; Keppel-Aleks et al., 2012; Zhao et al., 2016), and has the potential for confusion with other, covarying error sources (Basu et al., 2011). Among flux inversion systems that ingest in situ data, e.g., those analyzed by Gaubert et al. (2019), the spread in NBE is usually much smaller, about 0.25 Pg C. Nevertheless, flux inversions typically achieve this agreement through their reliance on long-term data records and transport simulations, and can trail the current date by a year or more. Our goal is to achieve a similar error reduction without the resulting latency, allowing the subsequent assimilation of satellite $CO_2$ data to focus on regional and seasonal errors rather than global budgeting errors.

This paper presents a collection of surface fluxes having both retrospective and forecasting modes that reproduce background $CO_2$ measurements with comparable skill to a modern flux inversion system. The collection includes an additional, empirically derived land sink to our baseline flux collection to ensure that global flux totals are consistent with observed atmospheric $CO_2$ growth rates. Empirical adjustments of this kind date back at least to the work of Tans et al. (1990), who showed that their simulations recreated observed North-South gradients of $CO_2$ when they increased terrestrial uptake in the Northern Hemisphere Extratropics. Later, Chevallier et al. (2009) tuned their fluxes to match observed atmospheric growth rates, Keppel-Aleks et al. (2012) adjusted Northern Hemisphere mid-latitude uptake to improve their simulated seasonal cycle amplitude, and Agustí-Panareda et al. (2016) derived an adjustment to fluxes from their prognostic model (i.e., one which does not ingest satellite vegetation data) based on comparisons to a flux inversion system. The method described here is an extension to that of Chevallier et al. (2009), while sharing some features of each of the previously cited works. In retrospective mode, it applies an atmospheric growth rate based on in situ observations in the marine boundary layer (MBL; Dlugokencky and Tans, 2016b). In forecasting mode, when many of the diagnostic flux products and observationally constrained growth rates are unavailable, the fluxes use extrapolation and a method for predicting the growth rate based on sea surface temperature forecasts (Jones and Cox, 2005; Betts et al., 2016). It differs from previous works in its functional form and its application to fluxes diagnosed from satellite measurements instead of those from a prognostic model. For now, we are willing to accept that the empirical sink adds a biophysical inconsistency into our system but hope to better address this in the future.

In the work that follows, the construction of the empirical sink is detailed in Section 2. The evaluation of the fluxes against other products and comparisons of transported mixing ratios to in situ measurements are presented in Section 3. In particular, we show that a GEOS simulation using the bias-corrected fluxes has similar skill reproducing MBL measurements to a simulation using fluxes from a modern inversion system. These findings and their potential scientific impact are summarized in Section 4.

## 2  The LoFI flux collection

Here we present the Low-order Flux Inversion (LoFI), a collection of carbon fluxes driven by remote-sensing land-surface data and bias-corrected to reproduce given atmospheric and oceanic growth rates. We use the term "low-order" to distinguish it from modern flux inversion systems which typically solve for fluxes at a regional, monthly scales or finer. To demonstrate the impact of the empirical sink, we also consider a "baseline" flux collection similar to that used in Ott et al. (2015) with the only difference being that the empirical land sink is removed. All products in both the LoFI and baseline collections are conservatively regridded to the $0.5° \times 0.625°$ dateline and pole centered grid used by GEOS for many applications including the MERRA-2 reanalysis of the meteorological inputs needed by the flux components (Section A1). All fluxes have a daily timestep, except the land biosphere which has a 3-hourly timestep to resolve the diurnal cycle. In retrospective mode, the LoFI collection trails the current date by up to a year. It forecasts fluxes for the current and next year using the modifications described in Section 2.3. As a rough metric of truth/plausibility, we compare both collections to widely-used ensembles of terrestrial biospheric, ocean biogeochemical models, and flux inversions. These choices and the components of the LoFI fluxes are detailed below and summarized in Table 1.

### 2.1  Individual flux components

The LoFI flux collection consists of the following six components:

**Net Ecosystem Exchange (NEE)** — An implementation of the Carnegie Ames Stanford Approach (CASA; Randerson et al., 1996) and Global Fire Emissions Dataset version 3 (GFED 3; van der Werf et al., 2003; 2010) referred to here as **CASA-GFED 3**. CASA-GFED 3 uses satellite-based measurements of land cover and vegetation changes along with meteorology from MERRA-2 to constrain carbon stocks and fluxes, viz. net primary productivity (NPP), which is determined from measurements of normalized difference vegetation index (NDVI; Pinzon and Tucker, 2014), and biomass burning, which is determined from Moderate Resolution Imaging Spectrometer (MODIS) burned area estimates (Giglio et al., 2010). This version is available on a $0.5$ degree grid with a monthly timestep. More details about our implementation and its use in GEOS are available in Ott et al. (2015).

**Biofuel** — **CASA-GFED 3** also produces an estimate of the anthropogenic burning of harvested wood (van der Werf et al., 2010), which we refer to here as biofuel. The emissions have no seasonality and are calculated as the population density times national per capita fuel consumption estimates while being constrained by the total available coarse woody debris at each model time step.

**Biomass burning** — The Quick Fire Emissions Dataset (**QFED**; Darmenov and da Silva, 2015), an NRT product, which determines emissions based on MODIS fire radiative power (FRP) estimates using a technique similar to the Global Fire Assimilation System (GFAS; Kaiser et al., 2012). QFED is produced on a $0.1$ degree grid for every day with a climatological diurnal cycle applied.

**Table 1.** The components of our fluxes and the different ensembles used for evaluation.

| Name | Type | References |
|------|------|-----------|
| *LoFI flux components* | | |
| CASA-GFED 3 | NEE & biofuel | van der Werf et al. (2003; 2010)[a] |
| QFED | Biomass burning | Darmenov and da Silva (2015) |
| ODIAC | Fossil fuel | Oda et al. (2018)[b] |
| LoFI Takahashi | Ocean | Section 3.1[c] |
| LoFI Empirical land sink | NEE adj. | Section 3.2[d] |
| *Top-down ensemble* | | |
| CarbonTracker 2016 & 2017 | NBE & Ocean | Peters et al. (2007)[e] |
| CarbonTracker Europe 2016 | NBE & Ocean | Peters et al. (2007)[f] |
| CAMS v17r1 | NBE & Ocean | Chevallier et al. (2011) |
| Jena CarboScope v4.1 S93 & S04 | NBE & Ocean | Rödenbeck et al. (2003) |
| *Bottom-up ensemble* | | |
| GCP 2018 (TRENDY V7) | NBE | Sitch et al. (2015)[g] |
| GCP 2018 | Ocean | Le Quéré et al. (2018)[h] |

[a] Implementation in GEOS described by Ott et al. (2015).

[b] Available at http://db.cger.nies.go.jp/dataset/ODIAC/.

[c] See https://www.esrl.noaa.gov/gmd/ccgg/carbontracker/CT2016/CT2016_doc.pdf for a more detailed description of a nearly identical approach used as an ocean prior in the NOAA CarbonTracker flux inversion system.

[d] See Chevallier et al. (2009) and Agustí-Panareda et al. (2016) for examples of similar approaches.

[e] Updates documented at http://carbontracker.noaa.gov.

[f] Updates documented by van der Laan-Luijkx et al. (2017).

[g] Updates documented by Le Quéré et al. (2018).

[h] Global, annual totals only.

**Fossil fuel combustion** — The Open-source Data Inventory for Anthropogenic $CO_2$ (**ODIAC**; Oda and Maksyutov, 2011, 2015; Oda et al., 2018). ODIAC is a global, monthly, high-resolution (1 km $\times$ 1 km) fossil fuel $CO_2$ gridded emission data product based on the disaggregation of country-level fossil fuel $CO_2$ emission estimates using a global power plant database and satellite observations of nighttime lights. It is updated on an annual basis upon the availability of updated global fuel statistical data. This work uses the 2016 version which covers 2000–2015 (the current version goes through 2019). For all but the two most recent years (here, 2014 and 2015), ODIAC uses global and country estimates from the Carbon Dioxide Information and Analysis Center (CDIAC; Boden et al., 2018), while estimates for the two most recent years are projected using BP's Statistical Review of World Energy 2016 (Oda et al., 2018).

125

**Ocean exchange** — An extension to the monthly climatology of Takahashi et al. (2009) that restores inter-annual variability. This approach reapplies the global mean climatological growth rate estimate of $1.5\ \mu\text{atm/yr}$ for the partial pressure of $CO_2$ in seawater ($\text{pCO}_2^{\text{sw}}$) that Takahasi et al. (2009) use to derive their climatology. For the partial pressure in the atmosphere ($\text{pCO}_2^{\text{atm}}$), we use weekly values of zonal-mean surface $CO_2$ from the NOAA MBL reference (Masarie and Tans, 1995; Dlugokencky and Tans, 2016a). Given the two partial pressures, estimates of the flux from the ocean surface to the atmosphere follow from the expression (Wanninkhof, 2014)

$$F^{\text{sw}} = kCU_{10}^2(\text{pCO}_2^{\text{sw}} - \text{pCO}_2^{\text{atm}}),$$

where $k$ is a constant, $U_{10}$ is the 10-meter wind speed, and $C$ is the fractional sea-ice coverage. To complete the flux calculation, we use daily, observationally-constrained estimates of $U_{10}$ and $C$ from MERRA-2. This approach has been used by several previous studies and is derived from one of the ocean priors in the NOAA CarbonTracker System (Table 1, footnote c).

**Empirical land sink** — An additional, empirical sink following the approach of Chevallier et al. (2009) that constrains the global atmospheric growth rate of the combined LoFI flux package. The empirical sink decreases heterotrophic respiration (HR) in months where the 2-meter air temperature ($T$), meant as a rough proxy for soil temperature, increases from the previous month. This is designed to concentrate the correction to the Northern Extratropics during the spring and summer, where the neutral biosphere assumption of CASA is thought to be most problematic (Yang et al., 2007; Carvalhais et al., 2008), and flux inversions indicate a net sink (Gaubert et al., 2019). For the $m$-th month of each year, at every point on the surface, the sink $\text{S}_m$ has the form

$$\text{S}_m = \alpha \cdot \Delta^+ T_m \cdot \text{HR}_m, \tag{1}$$

$$\Delta^+ T_m = \max(T_m - T_{m-1}, 0)/(10\ ^\circ\text{C}),$$

where $\Delta^+ T_m$ is the (non-dimensionalized) temperature increase from the previous month, and $\alpha$ is a constant scaling factor computed such that the global total fluxes for the year match a specified atmospheric growth rate $\Delta CO_2$. In other words,

$$\alpha = \frac{2.124 \cdot \Delta CO_2 - \langle F \rangle}{\langle \Delta^+ T \cdot \text{HR} \rangle},$$

where $2.124 \cdot \Delta CO_2$ is the atmospheric growth rate in units of Pg C (Ballantyne et al., 2012), $\langle X \rangle$ is the area-weighted global, annual total in units of Pg C of a flux field variable $X$, and $F$ is the sum of all baseline fluxes. In retrospective years (those preceding the current), we use growth rates derived from the NOAA MBL reference (Dlugokencky and Tans, 2016b), and in NRT years (the current and following) we use projections based on seasonal forecasts of sea surface temperature described in Section 2.3.

## 2.2 Anthropogenic short-cycle burning and lateral fluxes

The above separation into component fluxes assumes that, added together, biomass burning and biofuel emissions account for emissions from both naturally occurring wildfires and anthropogenic burning due to gross land-use/land-cover change. Fur-

thermore, emissions from ethanol, biodiesel, and other short-cycle fuels used for transportation are potentially underestimated since they are not included in the ODIAC fossil fuel and CASA-GFED 3 biofuel emissions, yet the removal of carbon due to the corn and soybean harvest in the Midwestern United States is included in CASA-GFED 3 (derived from USDA National Agricultural Statistics Service data for 2005). Excluding the lateral transport of significant amounts of carbon over continental scales, we do not expect these assumptions to have a noticeable effect on the comparisons to follow since they consider only NBE, the sum of all land fluxes except fossil fuel emissions, and not the individual components.

## 2.3  Modifications needed for forecasting mode

Many of the products used in the above fluxes are unavailable until a few months to years following the end of a given year. In particular, fossil fuel inventories and the NOAA MBL growth rate require data collection and analysis that must necessarily trail real time. Simulations during the current and future years are thus only possible with some type of extrapolation and/or statistical model applied to past values. For all flux components except biomass burning and the empirical sink, we produce estimates during NRT years by first extrapolating a linear fit through the retrospective values for each point and each month. This choice allows different regions and different seasons to have different trend lines. While more complex choices are possible, this is meant as a simple first step as we work to reduce the latency of our baseline satellite-derived flux products.

The primary motivation for using QFED biomass burning instead of GFED was QFED's ability to produce NRT estimates. This ability is particularly important for biomass burning emissions whose inter-annual variability is comparable in magnitude to its annual mean and seasonal cycle amplitude and is much greater than its long-term trend. While the removal of terrestrial carbon by QFED fires does not match the corresponding fire loss in CASA carbon stocks, QFED emissions are calibrated to GFED (Darmenov and da Silva, 2015), resulting in a difference that is minor compared to that of the empirical sink (see the Supplementary Section A3).

For the empirical sink, we switch from using an atmospheric growth rate derived from in situ data in retrospective years to a linear functional fit to seasonal SST anomalies and anthropogenic emissions (Jones and Cox, 2005; Betts et al., 2016). This approach estimates the growth rate of $CO_2$ in ppm, $\Delta CO_2$ as

$$\Delta CO_2 = 0.069 + 0.442N + 0.205E, \tag{2}$$

where $N$ is the average of SST anomalies in the Niño 3.4 region ($5\,°N$ to $5\,°S$, $170\,°W$ to $120\,°W$) from October of the previous year to September of the current year in units of degrees Kelvin, and $E$ is the global total anthropogenic emissions from fossil fuels and net land-use/land-cover change in units of Pg C. For the SST anomalies, we use the Reynolds analysis (Reynolds et al., 2007) until the last month it is available and fill in the remaining months with SST forecasts from the GEOS Subseasonal to Seasonal (S2S) forecast system (Molod et al., 2020), and for the anthropogenic emissions we use extrapolated totals from the Global Carbon Project (GCP) 2018 budget.

The coefficient $0.205$ multiplying the anthropogenic term in the growth rate forecast (Equation 2) represents a constant airborne fraction of $0.44$ when converted using a factor of $2.124$ Pg/ppm, roughly in line with the findings of Raupach et al. (2014). Historical data records tend to show that the long-term trend in the airborne fraction is insignificant compared to its

inter-annual variability (Knorr 2009; Ballantyne et al., 2012), but there is some indication of significant multi-decadal trends, including the possibility of a recent decrease (Keenan et al., 2016).

## 3   Flux and transport simulation analysis

Our primary method for evaluating the LoFI flux package, along with the baseline fluxes, are comparisons to the ensembles of bottom-up terrestrial biosphere and ocean biogeochemistry models and top-down flux inversions outlined in Table 1 and described in more detail in the Supplemental Section A2. It is important to note that the bottom-up ensembles and top-down ensemble are not directly comparable: riverine input of carbon from the terrestrial biosphere to the ocean causes the top-down ensemble to infer a greater terrestrial sink and smaller oceanic sink than the biospheric model ensemble (Le Quéré et al., 2018). Jacobson et al. (2007) used fluxes from a global erosion model (Amiotte Suchet and Probst, 1995; Ludwig et al., 1996) to estimate a riverine contribution of $0.45 \pm 0.18$ Pg C (all uncertainties reported here are $1\sigma$). Recently, Resplandy et al. (2018) showed that relationships between ocean heat and carbon transport to derive an estimate of $0.78 \pm 0.20$ Pg C. Since there is so much uncertainty about this discrepancy, in particular its distribution in time and space, we do not make any corrections to the ensemble ranges used in the comparisons, and this choice should be kept in mind in the interpretation of the following results. In any case, the most appropriate ensemble for our purposes is the top-down ensemble, while the bottom-up ensembles indicate a greater range of plausibility.

There are several other flux evaluation metrics, which we do not apply here, each with their own limitations. For example, fluxes can be measured directly from towers with eddy covariance techniques (Dabberdt et al., 1993), but the spatial footprint of a flux tower is typically much smaller than ten kilometers (Raczka et al., 2013), and upscaling flux tower data to a global, gridded product has thus far been unable to produce reliable global NBE budgets (Jung et al., 2011). Eddy covariance measurements from aircraft (Desjardins et al., 1989), e.g., NASA's Carbon Airborne Flux Experiment (CARAFE; Wolfe et al., 2018), are representative of much longer horizontal length scales than towers, but the currently available campaign data are sparse in space and time. As shown in Section 3.3, another choice is to transport our fluxes through the atmosphere with the GEOS general circulation model (GCM) and compare the simulated $CO_2$ mixing ratios with atmospheric observations. While this approach can test the ability of the fluxes to reproduce large-scale, long-term signals, it is susceptible to the misinterpretation of a transport error as a flux error.

### 3.1   Ocean exchange

Over the past few decades, global $pCO_2^{sw}$ has increased by roughly $1.5\ \mu\text{atm/yr}$, yet with considerable regional differences (Takahashi et al., 2009). At the same time, $pCO_2^{atm}$ has increased at an even greater pace, driving an increasing ocean sink. Deviations from these trends are predominantly limited to the Tropical Pacific, where the El Niño, Southern Oscillation (ENSO) is the dominant driver of interannual variability (Rödenbeck et al., 2015), and the Southern Ocean, where the net ocean sink switched from increasing to decreasing in the early 1990s and switched back in the early 2000s (Landschützer et al., 2015).

By imposing observed trends in $pCO_2^{sw}$ and $pCO_2^{atm}$, our ocean exchange fluxes produce a sink that is generally consistent with the inversion ensemble and GCP 2018 ocean biogeochemical model ensemble (Figures 1 and A1), again with the provision that the biogeochemical model ensemble does not include outgassing due to riverine input. Averaged over 2000–2010, our mean ocean sink is 1.55 Pg C. This is consistent with annual budgets based on atmospheric measurements of a combination of of $O_2$ and $CO_2$ called atmospheric potential oxygen (APO; Stephens et al., 1998): Keeling and Manning (2014) estimate an ocean sink of $2.50 \pm 0.60$ Pg C for 2000–2010, which reduces to $1.72 \pm 0.60$ Pg C after removing 0.78 Pg C to account for riverine input (see discussion above). The $pCO_2^{sw}$ based products of Landschützer et al. (2016) and Rödenbeck et al. (2014), which do not require a riverine adjustment, produce sinks of 1.37 and 1.74 Pg C (averages from GCP 2018; Le Quéré et al., 2018), also in line with our budgets (several more comparable estimates are available in Jacobson et al., 2007). Even during the strong ENSO of 2015–2016, the constant growth in $pCO_2^{sw}$ that we impose produces a global sink within the ranges of the top-down and bottom-up ensembles (Figure 1). While linear growth is not appropriate for simulations of several decades, over which $pCO_2^{sw}$ increases exponentially (Raupach et al., 2014), it is sufficient for our 15-year study period and something we hope to address in future developments.

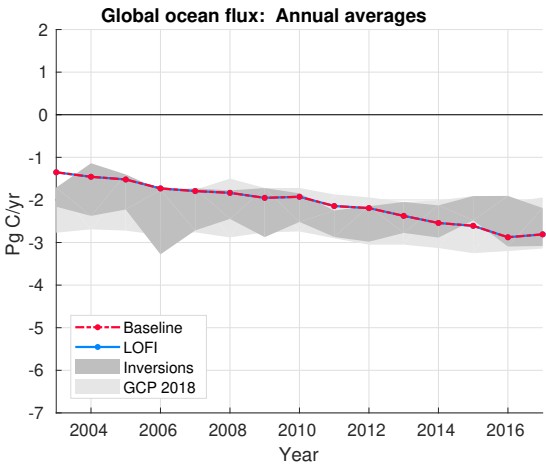

**Figure 1.** Global annual totals of LoFI ocean exchange fluxes (solid blue; identical to baseline) for 2003–2017. Min-to-max ranges of the inversion ensemble (dark grey) and GCP 2018 ocean biogeochemical model ensemble (light grey) are provided for comparison.

## 3.2 Net biospheric exchange

In comparison to the flux inversion ensemble and the TRENDY Version 7, Simulation 3 (Sitch et al., 2015) ensemble of dynamical global vegetation models (DGVMs), the baseline fluxes consistently underestimate NBE—the sum of NEE, biofuels, and biomass burning—by over 3 Pg C (see Figure 2a and Table 2). This is due to the assumption of a neutral biosphere in CASA-GFED, which thus has no long-term net sink, and is the primary motivation for the derivation of the empirical land sink. In the Northern Hemisphere, the neutral biosphere assumption causes CASA to systematically underpredict seasonal

cycle amplitudes in comparison to measurements from flux towers (Carvalhais et al., 2008) and a combination of aircraft and ground-based remote-sensing retrievals (Yang et al., 2007). This effect is seen in the diagnostic fluxes as an underprediction of the global sink strength in March through July (Figure 2) limited primarily to the Northern Extratropics (Figure 3, first column). In contrast, including the empirical sink in the LoFI fluxes moves its annual totals and seasonal cycle significantly closer to the ranges of the comparison ensembles. Since the empirical sink increases the seasonal cycle and its magnitude increases as net terrestrial uptake grows, it produces a seasonal cycle amplitude that increases in time, consistent with observations of in high northern latitudes (Graven et al., 2013) and their attribution to increased extratropical terrestrial uptake (Barnes et al., 2016).

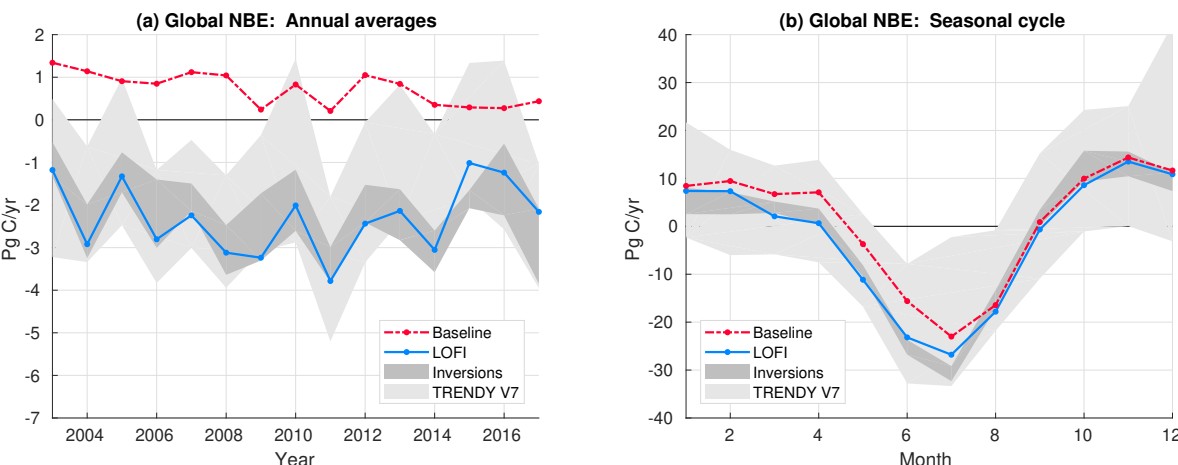

**Figure 2.** Global total NBE from fluxes excluding (baseline; dash-dot red) and including (LoFI; solid blue) the empirical sink: (a) annual averages and (b) the seasonal cycle climatology for 2003–2015. Min-to-max ranges of the inversion ensemble (dark grey) and TRENDY V7 ensemble (light grey) are provided for comparison.

Increasingly, modern flux inversions tend to predict a persistent, net sink in the Northern Extratropics (NE), with the Tropics and Southern Extratropics (T+SE) in near balance with considerably more inter-annual variability than the NE (Ciais et al., 2019; Gaubert et al., 2019). Stephens et al. (2007) first showed that inversions with neutral T+SE fluxes compared better to profiles of in situ aircraft data than inversions with a significant T+SE source. Inversions from the Regional Carbon Cycle Assessment and Processes (RECCAP; Canadell et al., 2011) intercomparison also showed a clear separation between neutral and source T+SE budgets, with the difference being due to the decision to average observational data over month-long intervals (Peylin et al., 2013). For the RECCAP inversions that used data varying within the month, Peylin et al. (2013) found a sink over 2001–2004 of $1.85 \pm 0.25$ Pg C in the NE and $-0.34 \pm 0.27$ Pg C in the T+SE. The agreement has become even stronger across more modern inversion systems, with those covering 2004–2014 finding a NE sink of $2.17 \pm 0.36$ Pg C and $0.06 \pm 0.11$ Pg C in the T+SE (Gaubert et al., 2019). Over this same time period, the LoFI fluxes have a NE sink of 2.50 Pg C and a T+SE sink of $-0.41$ Pg C, suggesting possibly a slight overestimation of the T+SE source, yet still within a reasonable range of uncertainty (see Table 2 for comparable ranges from the TRENDY and inversion ensembles).

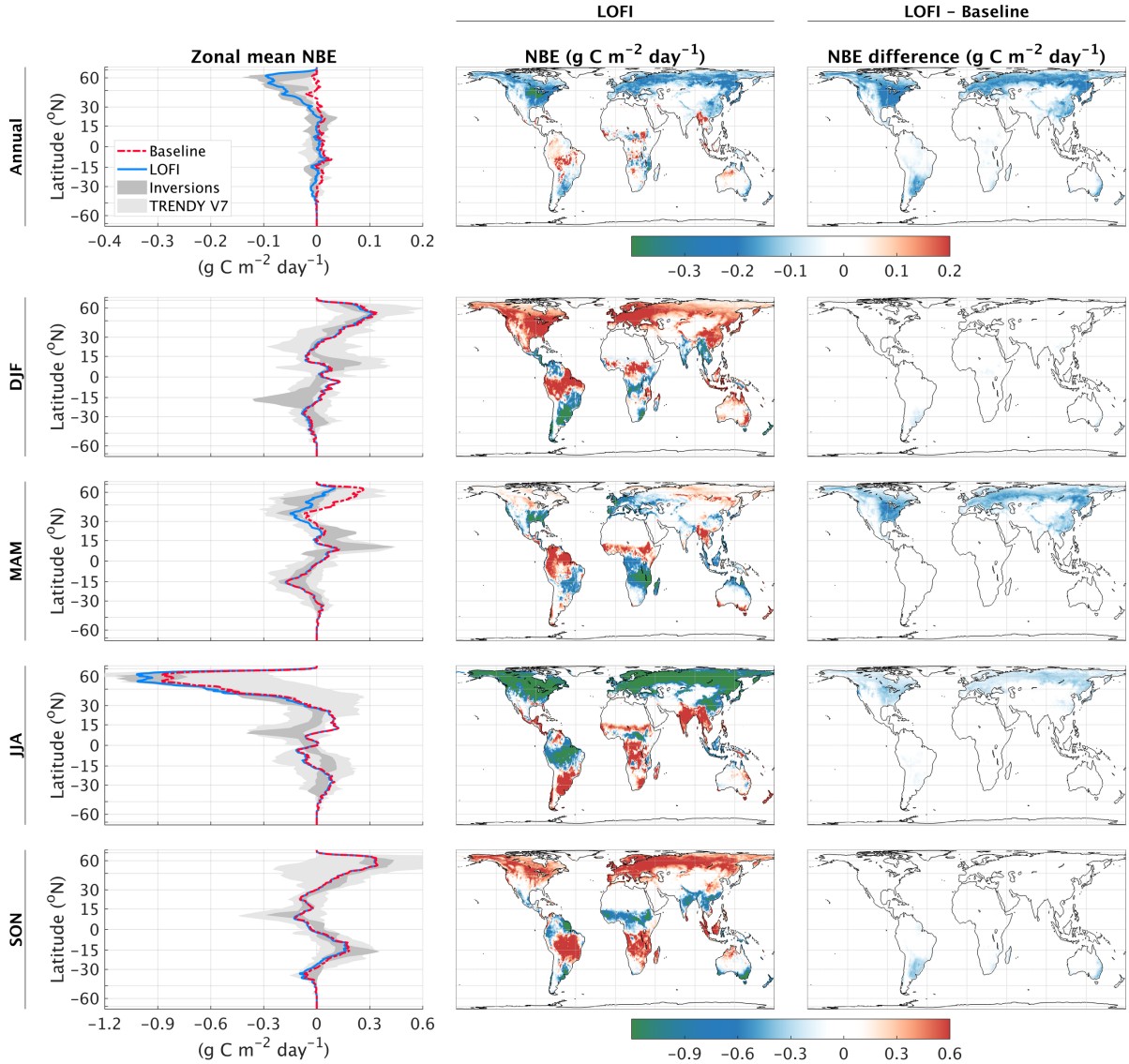

**Figure 3.** Climatologies (2003–2015) of NBE from the fluxes excluding (baseline) and including (LoFI) the empirical sink. The first row depicts annual averages and each subsequent row depicts averages over different seasons. In the left column, zonal mean NBE from the baseline fluxes (dash-dot red) and LoFI fluxes (solid blue) are plotted along with the ranges of the inversion ensemble (dark grey) and the TRENDY V7 ensemble (light grey); while the middle and right columns show maps of the LoFI fluxes (middle) and the magnitude of the empirical sink (right), i.e., the difference between the LoFI and baseline fluxes. Note that the scale of the annual average plots is three times smaller than that of the seasonal average plots.

**Table 2.** Comparison of 2004–2014 NBE budgets over the Northern Extratropics (NE) and Tropics and Southern Extratropics (T+SE).

| Product | NE (Pg C) | T+SE (Pg C) |
|---|---|---|
| Baseline | 0.07 | −1.24 |
| LoFI | 2.50 | −0.41 |
| Inversions[a] | 2.27 ± 0.36 | −0.10 ± 0.28 |
| TRENDY V7[a,b] | 1.27 ± 0.65 | 0.00 ± 0.48 |

[a] Ranges are the 2004–2014 average of the $1\sigma$ uncertainties across all products.
[b] Requires adjustment for riverine export, e.g., 0.78 Pg C added to the NE budget.

The net terrestrial sink found by inversions has been shown to be consistent with growth in temperate and boreal terrestrial ecosystems driven primarily by carbon fertilization and forest regrowth (Schimel et al., 2015; Fernández-Martinez et al., 2019)
with the possibility of an additional contribution from agriculture (Zeng et al., 2014). Notably, the forest inventory analysis of Pan et al. (2011) estimated a 2000–2007 sink of $1.28 \pm 0.17$ Pg C in boreal and temperate forests (mostly in the NE) and $-0.08 \pm 1.17$ Pg C in tropical forests. Further evidence for a persistent NE sink and a weak sink or source in the T+SE can be found in estimates of aboveground biomass change derived from vegetation optical depth (VOD; Liu et al., 2015), a product of remotely-sensed microwave radiation, and DGVM simulations constrained with forest demography data (Pugh et al., 2019),
the Global Forest Age Database (GFAD; Poulter et al., 2019). Pugh et al. (2019) find a significant sink in the Eastern United States, which is produced by our empirical sink as well (Figure 3, third row, third column). These studies do differ with Pan et al. (2011) in important ways, including the attribution of a greater percentage of the net tropical sink to gains in shrublands and savannas than to forest regrowth (Liu et al., 2015) and estimating a significantly smaller tropical regrowth sink (Pugh et al., 2019).

Taken together, the findings described above suggest an empirical sink proportional to the $CO_2$ growth rate (e.g., driven by carbon fertilization) and monthly temperature increase (e.g., focused to the extratropical growing season). The choice of the last remaining factor in Equation 1, heterotrophic respiration (HR) over, for example, Net Primary Production (NPP), requires further investigation. We chose HR rather than NPP for several reasons: 1) boreal forests allocate a much greater percentage of biomass below ground than tropical forests do (Pan et al., 2011), 2) the NDVI driver data of CASA tends to put a strong,
observationally-driven constraint on NPP, and 3) the uncertainty in the HR response to temperature changes through its so-called $Q_{10}$ function (Lloyd and Taylor, 1994; Tjoelker et al., 2001; Huntzinger et al., 2020). In any case, the temperature increase factor of the empirical sink makes it relatively insensitive to the choice of the flux factor (e.g., HR, NPP, or ecosystem respiration).

Even after adding the empirical sink, the LoFI fluxes have some discrepancies with the comparison ensembles worth noting.
In particular, the LoFI fluxes predict a sink from $0\,°S$ to $15\,°S$ during JJA, while the comparison ensembles predict neutral fluxes (Figure 3, first column, fourth row difference between blue line and grey shading). The opposite difference, although

less noticeable, is present during DJF (Figure 3, first, column, second row). It is unclear, however, how accurate either ensemble is in this case: the inversions are hindered by the limitation of in situ data and uncertainties in transport over this latitude band, while the biospheric models are potentially hindered by inaccuracies in meteorological driver data and deficiencies in the representation of disturbance (Molina et al., 2015; van der Laan-Luijkx et al., 2015). This approach may overestimate the net sink over the Midwestern United States, where our version of CASA-GFED 3 already includes a corn and soybean harvest (other harvests are not currently represented). Understanding the interaction of these two adjustments is the subject of ongoing work.

## 3.3 Transport simulations

Ultimately, a major goal of this effort is to develop a realistic collection of fluxes that improves the ability of transport simulations to reproduce measurements and support retrospective and NRT studies. As a test of this skill, we transport the LoFI fluxes through the atmosphere with the GEOS GCM and compare the results to in situ measurements from NOAA MBL sites available in NOAA ObsPack GlobalView+ v4.2.2 (Masarie et al., 2014; Cooperative Global Atmospheric Data Integration Project, 2019). We do the same with NOAA's CarbonTracker 2016 (CT2016; Peters et al., 2007, with updates documented at http://carbontracker.noaa.gov) fluxes for all components, which functions as a benchmark of the ability of the GEOS GCM to reproduce NOAA MBL measurements when using fluxes from a modern inversion system.

All transport simulations are run using the Heracles $4.0$ GEOS GCM version on a $0.5° \times 0.625°$ regular latitude-longitude grid with 72 vertical levels, a timestep of 15 minutes, and output instantaneous fields every 3 hours. The run uses the GEOS replay approach to reproduce the effect of the meteorological data assimilation system without having to rerun it (see Orbe et al., 2017 for the most up to date description). In this configuration, the large-scale circulation, temperature, and moisture are constrained by analysis fields every six hours, while physical processes such as convection, turbulence, and radiative transfer are recalculated at a high temporal resolution. This computationally efficient framework provides the ability to simulate realistic meteorology with a tight coupling between fine scale atmospheric transport processes and trace gas emissions. For the analyzed meteorology, we use MERRA-2, resulting in a transport simulation sharing many of the properties of that used by Ott et al. (2015).

The evaluation of the two model runs (one with LoFI fluxes, the other with CT2016 fluxes) against the NOAA MBL surface sites is shown in Figure 4. Neither of the runs is clearly superior. This suggests that, at least in terms of aggregate statistics over multiple years, the empirical sink produces a correction to the baseline, diagnostic fluxes with a similar skill as running a formal inversion system based on MBL data. While our approach may degrade the skill of CT2016 by using a different transport model, it may also improve it by running at a higher resolution, and our calculated differences are consistent with those in the evaluation in the CT2016 documentation. At this level of agreement with the surface sites, it is difficult to say if errors from atmospheric transport or any single component of the LoFI fluxes dominate all other sources of error. Further refinement of any single component then runs the risk of confusing errors in one component with those from another.

**Marine boundary layer OMF statistics**

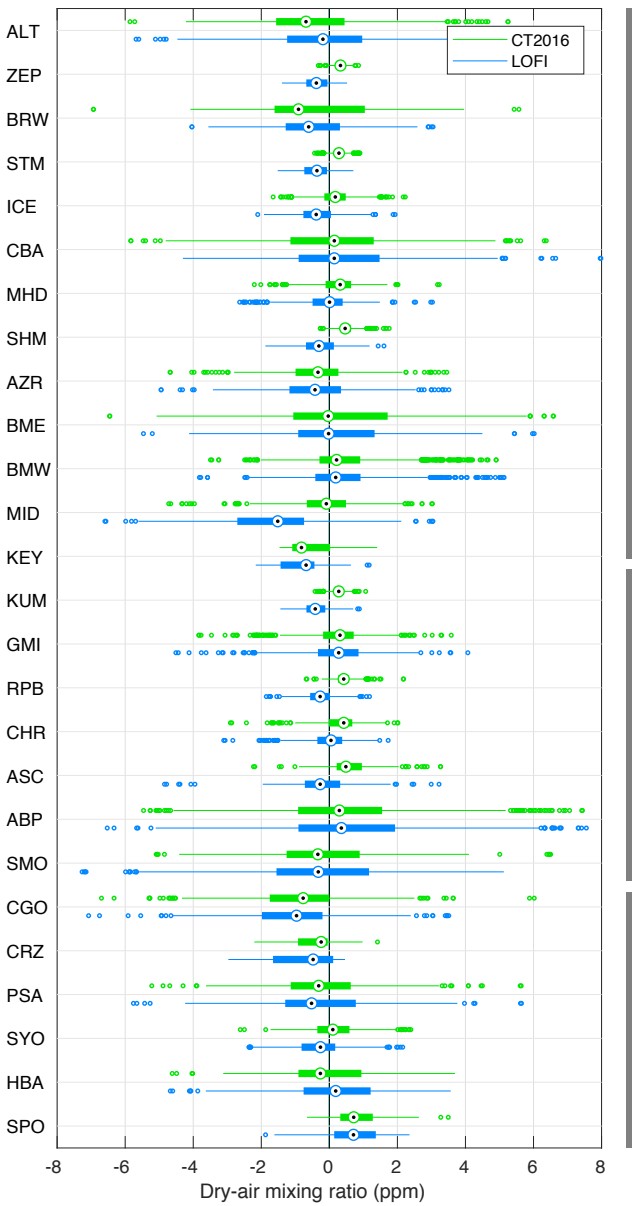

**Figure 4.** Box-and-whisker plots of observation minus forecast (OMF) statistics for 2003–2015 from model simulations using CT2016 fluxes (green) and using LoFI fluxes (blue). The comparison sites are all of the stations from the marine boundary layer collection and are ordered by latitude along the $y$-axis. For each site, the circular target denotes the median, the solid box denotes the range between the 25th and 75th percentiles, whiskers denote a range of roughly 99 percent of the data, and boxes denote values outside this range. Grey bars on the right indicate, from top to bottom, the Northern Extratropics (north of 23 °N), Tropics (between 23 °N and 23 °S, and Southern Extratropics (south of 23 °S).

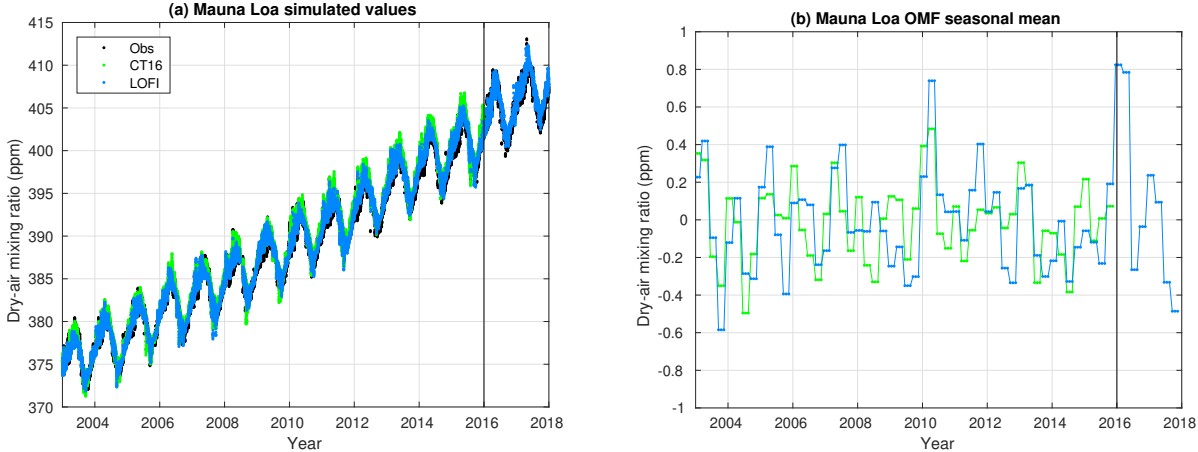

**Figure 5.** Time series comparisons of model simulations using CT2016 fluxes (green) and using LoFI fluxes (blue) to observations (black) at the Mauna Loa observatory. The left panel depicts all samples, while the right is a 3-month mean of the observation minus model differences. The solid black line at 2016 indicates the switch from retrospective to NRT modes in the LoFI fluxes.

### 3.4 Growth rate forecasts

At the start of 2016, the LoFI fluxes switch from retrospective mode to NRT. While it would be possible to extend retrospective mode to 2018 at the time of writing, we pick 2016 as an interesting test case where we must forecast a growth rate during a strong El Niño. In practice, NRT runs are limited to two years since the flux products in the LoFI collection have at most a two-year latency. For 2016 and 2017, Equation 2 predicts growth rates of 2.93 and 2.63 ppm, while the current NOAA MBL reference values are $2.85 \pm 0.09$ and $2.15 \pm 0.09$ ppm. These small differences do not appear to significantly impact the ability

of NRT LoFI model runs to reproduce in situ measurements at Mauna Loa (Figure 5)—while there is a notable error near the start of 2016, it is comparable in size to errors during the 2010–2011 El Niño and all errors after 2016 stay below 1 ppm in magnitude. This suggests that misrepresentations of the spatial and temporal variability of El Niño may have a greater impact on our ability to represent the Mauna Loa data record than errors in the growth rate forecast.

  A major factor in the ability of NRT LoFI to reproduce atmospheric $CO_2$ observations is that, even in forecasting mode, it

continues to use the retrospective meteorological analysis from MERRA-2. If we were to use meteorological forecasts instead, we would expect the skill to degrade within a few months and last about two years or less (Ilyina et al., 2020).

### 4   Conclusions

This paper presented an adjustment to a diagnostic collection of surface fluxes designed to bias-correct its global budgets to match inventory data and in situ observations. For the adjustment, we developed an empirical sink that was the product of

the monthly increase in temperature with heterotrophic respiration, thus focusing the correction to the Northern Extratropics

(NE) in spring and summer. The seasonal and zonal variability of the bias-corrected fluxes were shown to be consistent with ensembles of flux inversions and DGVMs and a growing body of literature indicating a net sink in the NE (Gaubert et al., 2019) driving increases in seasonal cycle amplitude in the high northern latitudes (Graven et al., 2013; Barnes et al., 2016). This suggests that diagnostic models of the terrestrial biosphere have the potential to reproduce these patterns with the addition of a transient carbon pool representing the net sink in the NE.

Using the fluxes bias-corrected with the empirical sink in transport simulations reproduced atmospheric measurements in the marine boundary layer with the same skill as transport simulations using fluxes from a flux inversion system. In particular, the annual total errors were consistently less than a ppm, with the true value almost always falling within a quartile of the differences. Globally, the errors are on the order of a few tenths of a ppm. The empirical sink thus enables the study of carbon cycle anomalies, like the 2015–2016 El Niño, whose effect on the atmospheric growth rate is a few ppm or less. By removing the dominant error contribution to our baseline fluxes, the lack of a net sink, the bias-correction also opens the possibility of correcting for errors in other flux components, e.g., fossil fuel emissions, a major, long-term scientific goal needed in the implementation and verification of international climate accords.

There are several benefits to using this approach to bias-correct a system's surface fluxes. In comparison to a flux inversion system, it is exceedingly simple to implement. It is also less susceptible to errors due to the particular transport model, data selection, and error covariance models. For example, constraining the global growth rate alone requires few, if any, assumptions about atmospheric transport or decorrelation times and lengths. When used in simulations or as a prior in an assimilation system, this approach significantly reduces biases due to misspecification of the growth rate. Failing to remove such a bias before assimilating data limits the ability of the assimilation system to account for other signals of interest in the observations (Dee, 2005), e.g., synoptic-scale variations due to passing weather systems, regional and seasonal anomalies due to drought, and changes in anthropogenic emissions. Finally, our approach produces high-resolution fluxes which are often precluded by the computational demands of flux inversion systems.

Since observational estimates of the global growth rate are currently only available at the end of each year, using the empirical sink developed here in an NRT atmospheric monitoring system requires a prediction of the global growth rate. We projected growth rates in 2016 and 2017 based on forecasts and analyses of SST (Jones and Cox, 2005; Betts et al., 2016). The values of 2.93 and 2.63 ppm were reasonable estimates of the values measured in the MBL of $2.85 \pm 0.09$ and $2.15 \pm 0.09$ ppm which are unavailable until a few months after the year's end. The predicted $CO_2$ mixing ratios showed comparable skill in reproducing in situ observations. Combined with the future ability to assimilate satellite retrievals of $CO_2$ lagging real time by just a few days, we expect to be able to monitor and predict growth rates in NRT.

*Data availability.* The LoFI fluxes are available upon request from the corresponding author. NOAA CarbonTracker, CarbonTracker Europe, and the CAMS flux inversions are all publicly available from their institution's websites. All other data products used in this work (viz., the TRENDY ensemble and the Jena CarboScope flux inversion) must be obtained from the respective project leads.

## Appendix A: Supplementary material

### A1    The downscaling algorithm

Because of the rectifier effect, if a model transport simulation hopes to reproduce the observed latitudinal gradient, it must use surface fluxes with a diurnal cycle (Denning et al., 1995). To resolve the diurnal cycle of the fluxes, we first downscale the monthly LoFI fluxes to daily using the the algorithm described below. We stop at daily for all fluxes other than terrestrial NEE, which is the difference of ecosystem respiration (ER) and GPP. Daily terrestrial ER and GPP are downscaled to 3-hourly following the approach of Olsen and Randerson (2004) with the slight modification of starting from daily instead of monthly
fluxes. This approach is quite similar to the downscaling approach used by NOAA's CarbonTracker system. It has the advantage of avoiding the noticeable discontinuities at monthly boundaries that are present in the monthly to 3-hourly downscaling, but the disadvantage of possibly missing synoptic scale disturbances that occur over multiple days since the downscaling from monthly to daily uses interpolation.

Fluxes are downscaled to a higher resolution either spatially or temporally by finding the smoothest interpolant that preserves
the averages at the original, coarser resolution. This interpolant is found by minimizing a quadratic cost function subject to a linear constraint. The quadratic cost function is the square of the discrete approximation to the Laplacian of the downscaled field. When interpolating in space, we use the spherical Laplacian, which takes the shrinking distance between grid boxes near the poles into account. The linear constraint is that the averages of the downscaled field at the coarser resolution be the original values. It is imposed using Lagrange multiplier, which transforms the problem into an unconstrained quadratic optimization
problem on a higher-dimensional space. This approach is used to downscale monthly NEE to daily as described above and is applied and to downscale ocean $pCO_2^{sw}$ from its native $4° \times 5°$ spatial grid to the $0.5° \times 0.625°$ grid of MERRA-2 (as shown in Figure A1).

### A2    The evaluation ensembles: further details

We evaluate our fluxes using a top-down ensemble of modern flux inversion systems and bottom-up ensembles of terrestrial
biosphere and ocean biogeochemical models. The flux inversion ensemble consists of the 2016 and 2017 versions of NOAA CarbonTracker (CT2016 and CT2017; Peters et al., 2007, with updates documented at http://carbontracker.noaa.gov), Carbon-Tracker Europe (CTE) version 2016 (van der Laan-Luijkx et al., 2017), Copernicus Atmospheric Monitoring Service (CAMS) version 17r1 (Chevallier et al., 2011), and the S93 and S04 runs of Jena-CarboScope (JCS) version 4.1 (Rödenbeck et al., 2003). The bottom-up model ensembles are the same as those used in the Global Carbon Project, 2018 (GCP 2018; Le Quéré
et al., 2018): the terrestrial model ensemble is all TRENDY Version 7, Simulation 3 (Sitch et al., 2015) dynamical global vegetation models except LPJ-GUESS, which did not submit monthly results, and the ocean model ensemble is the collection of global annual ocean exchange totals reported in GCP 2018. While comparison to these ensembles is not a true validation of our fluxes and is susceptible to uncertainties in lateral exchanges between the land and ocean, we do expect it to indicate when and where our surface flux product is an outlier compared to other estimates. In particular, we use the ensembles to identify
coherent, systematic surface flux errors over wide zonal bands and multiple months.

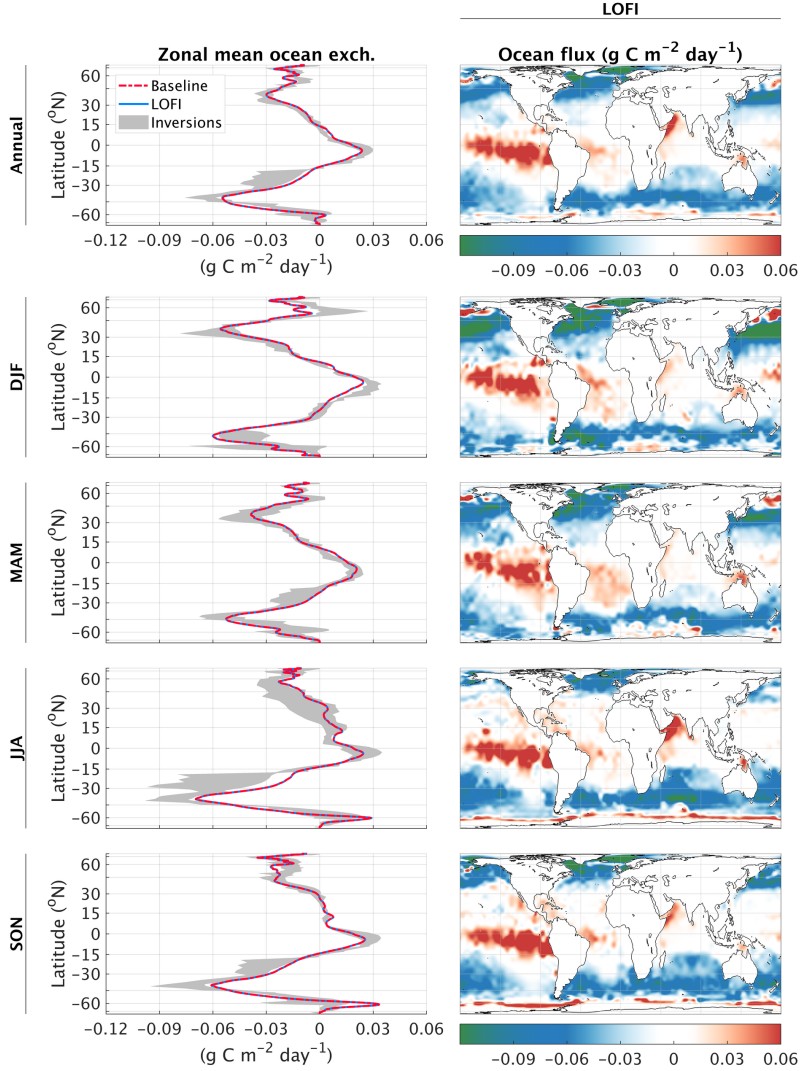

**Figure A1.** Climatologies (2003–2015) of ocean exchange from the LoFI fluxes. The first row represents annual averages and each subsequent row represents averages over different seasons. The left column depicts zonal means from the LoFI fluxes (dash-dot red) and the ranges of zonal means from the inversion ensemble (dark grey); and the right column shows seasonal gridded maps of the diagnostic fluxes.

The ensemble comparisons only consider net biospheric exchange (NBE), the sum of net ecosystem exchange (NEE), biomass burning emissions, and all other emissions not due to the combustion of fossil fuels. For the inversion ensemble, we compute NBE by subtracting a common fossil fuel product, which we use in our diagnostic fluxes, from the total surface flux. No distinction is made here between changes in NEE and biomass burning due to natural effects or to anthropogenic activities such as land use change. To avoid issues associated with statistics on small sample sizes or bias due to over representation of certain model configurations, we use only the range of the minimum and maximum values over each ensemble.

## A3  QFED versus GFED in LoFI

The main motivation behind replacing GFED in CASA-GFED with QFED is to avoid having to extrapolate biomass burning emissions in the forward processing product. Unlike other flux components, we do not expect the construction of a month-by-month linear climatology to have much skill for biomass burning. Rather than switching from GFED to QFED in the switch from reanalysis to forward processing fluxes, we chose to always use QFED. This prevents the introduction of a jump in the fluxes and allows a more direct comparison of averages but has the downside of introducing a biomass burning component that is not in balance with the carbon stocks of CASA. In any case, as is shown in Figures A2 and A3, the difference between QFED and GFED biomass burning emissions is quite small at almost every time and place. The only noticeable difference is that the baseline fluxes are slightly lower in the Amazon and Congo rain forests during JJA. Still, this difference is minor in comparison to the empirical sink.

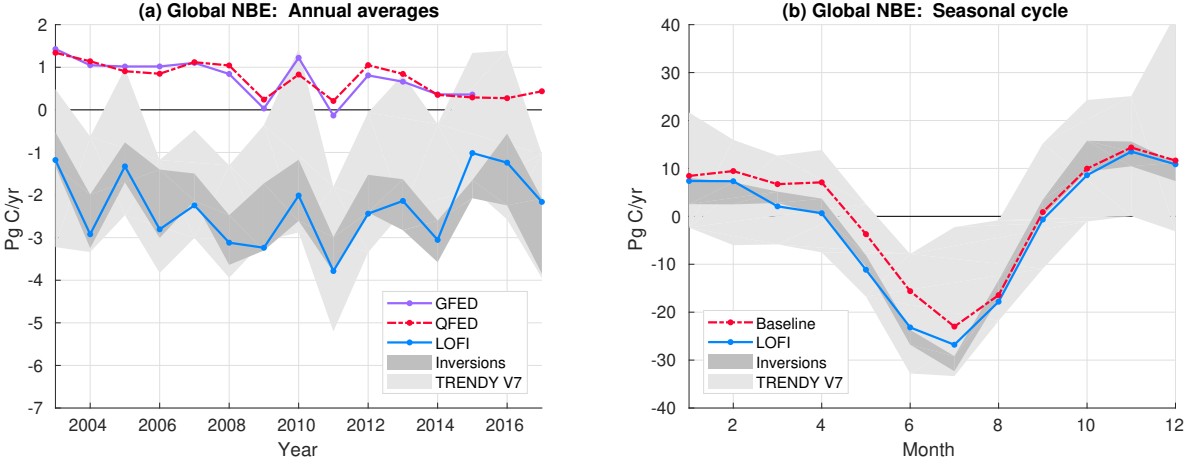

**Figure A2.** Global total NBE from the fluxes using GFED (solid purple) and QFED (dash-dot red) biomass burning: (a) annual averages and (b) the seasonal cycle climatology for 2003–2015. Min-to-max ranges of the inversion ensemble (dark grey) and TRENDY V7 ensemble (light grey) are provided for comparisons.

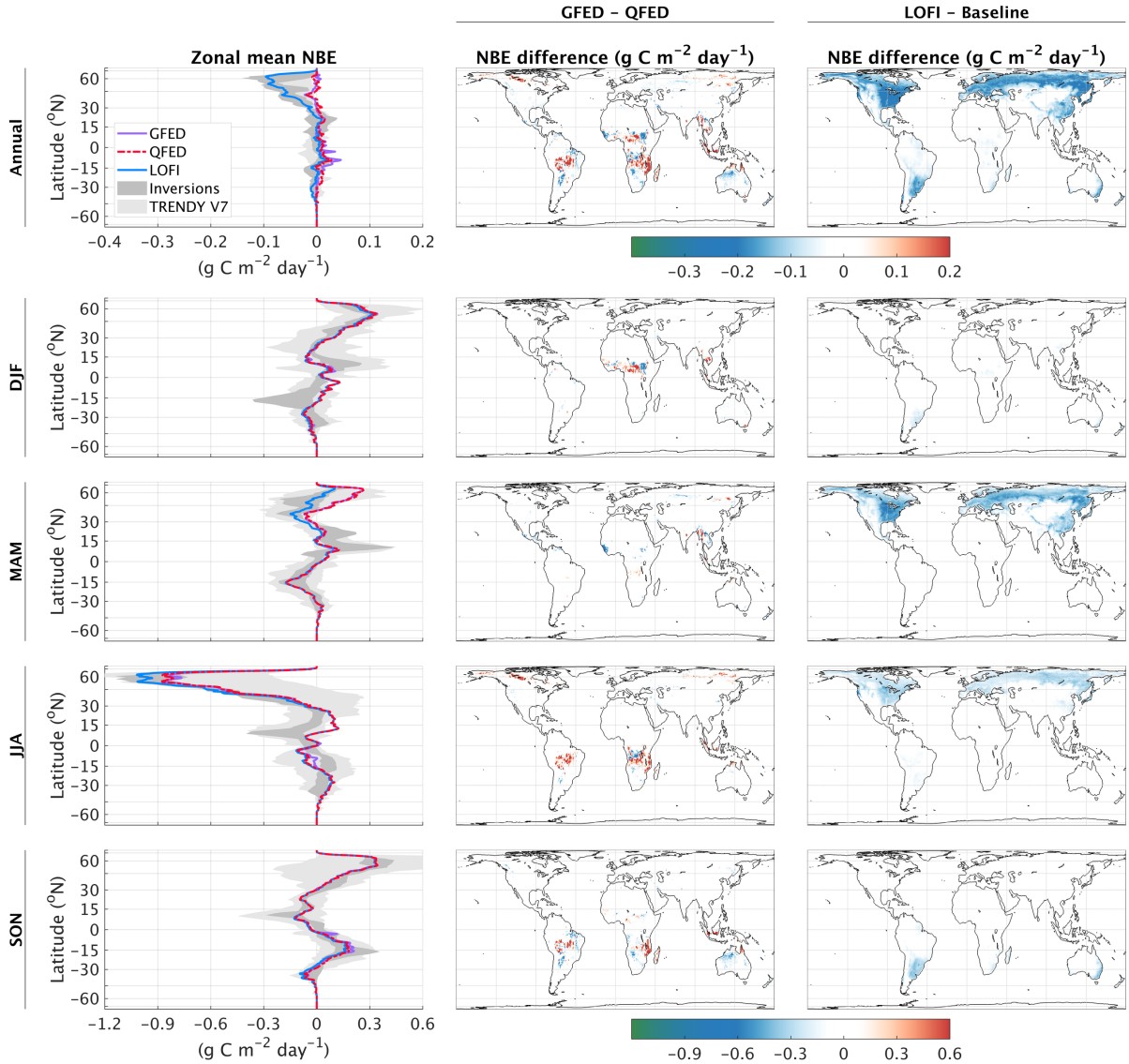

**Figure A3.** Climatologies (2003–2015) of NBE from the fluxes using GFED (solid purple) and QFED (dash-dot red) biomass burning and the LoFI fluxes (solid blue) for comparison. The first row represents the annual total and each subsequent row represents a different season. The left column depicts zonal means of the baseline fluxes using GFED and QFED, the range of the inversion ensemble (dark grey), and range of the TRENDY V7 ensemble (light grey); the middle column depicts the distance of the baseline fluxes using GFED to the range of the inversion ensemble; and the right column the difference of the baseline fluxes using QFED with those using GFED, i.e., the difference of QFED with GFED.

*Author contributions.* All authors contributed to the development of the ideas within and the composition of the manuscript. George J Collatz develops and maintains the CASA model and Tomohiro Oda develops and maintains the ODIAC product.

*Competing interests.* No competing interests are present.

*Acknowledgements.* This work was supported by funding from the NASA Carbon Monitoring System (NNH16ZDA001N). All simulations were performed on the Discover supercomputer at the NASA Center for Climate Simulation (NCCS). The authors would like to thank the Global Carbon Project and TRENDY intercomparison project as well as Stephen Sitch, Pierre Friedlingstein, and Masayuki Kondo in particular for their assistance with the interpretation and analysis of the TRENDY model outputs. We would also like to thank Frederic Chevallier, Andrew Jacobson, and Christian Rödenbeck for their assistance with the flux inversion products in this paper, and Sourish
Basu, David Baker, John Miller, and Britton Stephens for their comments and suggestions. Data from the 2017 Global Carbon Budget is archived at https://www.globalcarbonproject.org. TRENDY model output was provided by request by Pierre Friedlingstein and Stephen Sitch. NOAA CarbonTracker CT2016 and CT2017 results were provided by NOAA ESRL, Boulder, Colorado, USA from the website at http://carbontracker.noaa.gov. CarbonTracker Europe results provided by Wageningen University in collaboration with the ObsPack partners (http://www.carbontracker.eu). CAMS surface fluxes are produced at the Climate and Environment Sciences Laboratory (Laboratoire des Sciences
du Climat et de l'Environment; LSCE) and are available at https://apps.ecmwf.int/datasets/data/cams-ghg-inversions/. Jena CarboScope were provided by Christian Rödenbeck at the Max Planck Institute for Biogeochemistry and downloaded from http://www.bgc-jena.mpg.de/CarboScope/.

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
