# Peer review of "Bias-correcting carbon fluxes derived from land-surface satellite data for retrospective and near real-time assimilation systems"

_Atmospheric Chemistry and Physics, 2020_

## Referee Comment (RC1) · Anonymous Referee #1 · 30 Sep 2020

This manuscript describes a new system for near real time analysis and forecast of global carbon fluxes. The aim is to allow a fast analysis of the actual state of the global carbon cycle in support of satellite data evaluation, allow for a rapid response to newly observed anomalies, prepare for targeted measurement campaigns, provide a reference for extended assimilation of data, etc. The first results indicate that the performance that is achieved is comparable to state-of-the-art inversions. In my opinion this is a rather sobering outcome, putting the inversion community with their feet on the ground about what can be achieved. But I wonder also if it is fair given the focus on global or long-term mean fluxes in the performance evaluation. Furthermore, it is unclear whether the presented evaluation addresses the requirements of

the system given its objectives. Without a specification of those requirements from the start it is very hard the judge how well the system is supposed to perform. Currently, the implicit assumption seems to be that it shouldn't perform significantly worse than state-of-the-art inversions and global and climatological means, however, without further quantification. The structure of the manuscript is a strange mix of method, results, and discussion. I found myself going backwards and forwards to make sure that I read all the parts necessary to understand what was done. Furthermore, I didn't find any clear conclusions in the conclusion section. From this I conclude that the purpose of the paper is mainly to document the first stage of the NRT data assimilation system, for which a journal like GMD would have been more appropriate.

SPECIFIC COMMENTS

Title: I have difficulty with the word 'calibrating' here. The suggestion is made the method calibrates satellite measurements, which is really not what is done. Maybe something like 'bias-correcting' would solve this problem.

2 The LoFi flux collection: The structure of this section is unclear to me. I had expected three sections, one for the 'retrospective mode' on for the 'forecasting mode' preceded by everything that is common for these modes. I thought the latter was the baseline, which confusingly enough is not exactly what it turns out to be (see my next point).

Line 91: If I understand well the baseline still requires the NOAA MBL CO2 measurements for the ocean flux, which would make it a "retrospective" type analysis. Some explanation is needed of the purpose of the 'baseline' other than the notion that it doesn't include the empirical land sink. Initially I was assuming that it would be independent of the NOAA MBL CO2 measurements, which apparently is not the case.

Line 110 'Biofuel' and 'Biomass burning': What prevents double counting when combining these components?

Line 124: 'Estimates for the two . . . Review of World Energy 2016' How is this done, by

country, energy sector, or both?

Line 135: 'More information is available in Sections 3.1 & A3': For the method, not really. Those sections point to evaluation results. In the case of A3 only a single sentence is about the ocean, which could easily have been included in section 3.1.

Line 139: 'This is designed ... spring and summer' What is the design? Is equation 1 only applied to the northern extra-tropics? Per model grid box? What is the spatio-temporal discretization of $\alpha$? If it is only applied to the northern extra-tropics than what justifies the assumption that the residual land sink in CASA is fine elsewhere? Further details are needed here.

Line 148: 'about the construction and evaluation of the empirical sink, see Sections 3.2 & A3' Fine to put details in A3 (even though I only found information about the ocean and biomass burning there), but evaluation section 3.2 should not deal with the construction of the empirical land sink.

Line 152: ', yet the sink due to the corn and soybean harvest ....' This suggests that the midwest is the sink accounts for the global emission of short cycle fuels.

Line 176: 'in the Niño 3.4 region' Either this region needs to be defined, or a reference should be given where this information can be found.

Line 215: 'our ocean exchange fluxes produce a sink that is generally consistent with the inversion ensemble' Whether or not this result is consistent enough depends on the requirements. I would agree that the average sink is in good agreement, however, the trend is not. There is no discussion whether or not this is important, but it seems that a NRT projection or forecast would quickly divert from the uncertainty range.

Section 3.2 The empirical land sink: According to the components specified in section 2 this does not include biomass burning and biofuel. Yet in the description of this section numbers are provided for NBE. This should be made consistent.

Line 230: 'the sum the sum'

Line 267: 'adjustments to HR or NPP are both ... we look for in the empirical sink.' Given this conclusion from the preceding discussion, what is it that justifies the current treatment of the empirical land sink?

Line 294: 'This suggests that ... diagnostic fluxes with a similar skill as running a formal inversion system based on MBL data.' I do not agree with this for two reasons: 1) the agreement between CT2016 and NOAA MBL sites would have been much better when using its native transport model, 2) Table 1 suggests that only MBE and Ocean fluxes from CT2016 are used. If these are combined with different anthropogenic fluxes then this would add further inconsistency. It would have been fairer to use the CarbonTracker optimized concentrations in this comparison. In particular, because the empirical land sink didn't suffer from the same transport inconsistency.

3.4 Growth rate forecast: The authors indicate themselves that they could have extended the retrospective mode until 2018. It is not clear why this has not been done. It would have significantly strengthened the evaluation of the skill of the NRT mode (I mean by doing both modes for the 2016 – 2018 time window).

Line 362: 'When necessary, fluxes are downscaled to a higher resolution' It is not clear if this is done, or whether it is only a general possibility. It is also unclear which fluxes would require this step. If it is not used in the current setup then I recommend deleting this part.

---

## Referee Comment (RC2) · Anonymous Referee #2 · 16 Oct 2020

**Review:** Calibrating satellite-driven carbon fluxes for retrospective and near real-time assimilation systems, by Weir et al.

**Summary:** The authors describe a CO2 surface flux product that operates in retrospective and forecast modes, and can be provided with short latency. The product is shown to have comparable skill to full flux inversions that take much longer to calculate. Flux components such as biofuel, biomass burning, fossil fuel, and ocean flux are taken from near real-time or published datasets. Terrestrial carbon flux is obtained from CASA-GFED; the innovation, or 'hook' here is the imposition of an empirical land sink that concentrates the terrestrial sink in the northern extratropics during spring and summer, which is consistent with emerging inversion results (as opposed to placing the sink in the tropics or southern extratropics). This product, LoFI, is a suitable 'prior' for inversion studies, as it has total surface flux (and spatiotemporal distribution of flux) comparable to optimized fluxes from recent inversion projects.

**Review:** This is a good paper. It is concise and well-written, and the product it describes has value to the scientific community. I've had the opportunity to see LOFI results presented in meetings and workshops, and I recognize this value. ***My formal recommendation is to accept the manuscript for publication, with minor revisions.***

Initially I was a little concerned with the empirical sink, as it seemingly violates some accepted aspects of biophysics. After some thought, however, I realize that the authors are less concerned with maintaining fidelity to established physical relationships than they are with maintaining fidelity with flux inversion results, which do not take the physics into account. Inversions just say "here is what we think the flux map looks like".

Take heterotrophic respiration. There is a rich body of literature that describes how respiration increases with increasing temperature, and this is the basis for the so-called 'Q10' relationships present in just about every model that simulates surface CO2 flux (CASA as well, I believe). In LoFI, they *reduce* respiration as temperature increases. As stated previously, I think this is tolerable because making this assumption produces the flux map that they want. However, I think the authors need to acknowledge that this assumption violates accepted biophysical theory.

I'm also a little concerned about the strong MAM uptake in the Midwest crop region, shown in the third row of Figure 3. Yin et al. (2020), in a paper describing carbon uptake delay induced by floods in 2019, show (their Figure 2) show that in most years crops aren't even planted until April or May. It's hard to believe that these regions would show a significant sink immediately after plant date.

Again, I'm ok with this as long as the authors acknowledge that they are trying to reproduce the maps suggested by inversions, and not doing so with a strong regard for biophysical processes. I'd like them to find a way to say "Hey, we don't care about the biophysics. The inversions tell

us this is the pattern we want to have, and this is how we get it." I think this admission is important.

Other than that, I don't have much to add. Good paper, nice read, valuable product. Good job, wish all my reviews were this easy.

**Specific Comments:**

Lines 152-153: The Midwest crop harvest is not a true sink. They don't take the harvest and bury it deep in the ground. The harvest is respired back, from feedlots and from people who eat food made from the harvest. This must be accounted for in models. What does CASA do about this?

Line 177: Where does the land cover change map come from?

Line 212: I like the umlaut in El Nino.

Figure 1: It is hard to see LoFI in this plot. Is it directly under baseline? If the lines were thicker and the shading lighter, it would be easier to see. The scale can be shrunk too.

Line 230: typo

Lines 269-270: Boy, that discrepancy is really hard to see. Can you give us a number in the sentence describing it?

Figure 4: A line at the equator might be helpful, to show how many stations are in the Northern Hemisphere, and how many in the south. For those of us that don't have all the stations memorized yet.

Figure A3: It is really hard to see GFED. I really had to work my old eyes to see the tropical JJA difference.

**Reference:**
Yin, Y., Byrne, B., Liu, J., Wennberg, P., Davis, K. J., Magney, T., et al. (2020). Cropland carbon uptake delayed and reduced by 2019 Midwest floods. AGU Advances, 1, e2019AV000140. https://doi.org/10.1029/2019AV000140

---

## Author Comment (AC1) · 17 Nov 2020

Reviewer comments in red, responses in black.

This manuscript describes a new system for near real time analysis and forecast of global carbon fluxes. The aim is to allow a fast analysis of the actual state of the global carbon cycle in support of satellite data evaluation, allow for a rapid response to newly observed anomalies, prepare for targeted measurement campaigns, provide a reference for extended assimilation of data, etc. The first results indicate that the performance that is achieved is comparable to state-of-the-art inversions. In my opinion this is a rather sobering outcome, putting the inversion community with their feet

on the ground about what can be achieved. But I wonder also if it is fair given the focus on global or long-term mean fluxes in the performance evaluation. Furthermore, it is unclear whether the presented evaluation addresses the requirements of the system given its objectives. Without a specification of those requirements from the start it is very hard the judge how well the system is supposed to perform. Currently, the implicit assumption seems to be that it shouldn't perform significantly worse than state-of-the-art inversions and global and climatological means, however, without further quantification. The structure of the manuscript is a strange mix of method, results, and discussion. I found myself going backwards and forwards to make sure that I read all the parts necessary to understand what was done. Furthermore, I didn't find any clear conclusions in the conclusion section. From this I conclude that the purpose of the paper is mainly to document the first stage of the NRT data assimilation system, for which a journal like GMD would have been more appropriate.

The authors feel this work is well-suited to ACP since the journal has a long history of publishing important new results about carbon dioxide surface flux inversions. That literature forms the backbone of the findings in our research. As the reviewer notes, that LoFI performs so well compared to a modern flux inversion is a "sobering outcome, putting the inversion community with their feet on the ground about what can be achieved." In particular, this paper shows that adding a simple Northern Extratropical land sink to our a priori, "baseline" fluxes is enough to reproduce most of the skill of a modern flux inversion. We feel that this alone is an important scientific finding that would be outside the scope of a journal like GMD despite its impressive collection of Earth system model development literature.

SPECIFIC COMMENTS

Title: I have difficulty with the word "calibrating" here. The suggestion is made the method calibrates satellite measurements, which is really not what is done. Maybe something like "bias-correcting" would solve this problem.

The authors agree that the terminology is the title is imperfect, but feel it is technically appropriate. The use of "bias-correcting" may suggest to the reader that our correction is constant in time, while in fact it affects the inter-annual variability of the fluxes as well. In light of these considerations, we do not see a strong justification for changing the title, especially since keeping the current title allows for greater traceability.

2 The LoFi flux collection: The structure of this section is unclear to me. I had expected three sections, one for the "retrospective mode" on for the "forecasting mode" preceded by everything that is common for these modes. I thought the latter was the baseline, which confusingly enough is not exactly what it turns out to be (see my next point).

The first part of the section describes the components of the flux collection, and the second describes what we have to do differently in NRT. The baseline is simply the LoFI flux collection without the empirical land sink, as is noted in the paper on Line 91.

Line 91: If I understand well the baseline still requires the NOAA MBL CO2 measurements for the ocean flux, which would make it a "retrospective" type analysis. Some explanation is needed of the purpose of the "baseline" other than the notion that it doesn't include the empirical land sink. Initially I was assuming that it would be independent of the NOAA MBL CO2 measurements, which apparently is not the case.

You do understand that correctly. The main purpose of the paper was to evaluate the empirical land sink, so we kept all other products the same in the baseline package. While it would be interesting to test against an alternative baseline that does not use NOAA MBL CO2 measurements in any way, testing the importance of the MBL CO2 measurements, especially for constructing ocean fluxes, was not a primary objective of this paper.

Line 110 "Biofuel" and "Biomass burning": What prevents double counting when combining these components?

This is addressed in the paragraph immediately following the description of the components. In the original manuscript it beings on line 149.

Line 124: "Estimates for the two ... Review of World Energy 2016" How is this done, by country, energy sector, or both?

This is done by country and fuel type. ODIAC, like CDIAC, is a fuel-based (e.g., oil, coal, etc.), not sector-based inventory. This is part of the ODIAC product that we use as an input and described in great detail in Oda et al. (2018). We've slightly restructured the two sentences here to make this clearer.

Line 135: "More information is available in Sections 3.1 A3": For the method, not really. Those sections point to evaluation results. In the case of A3 only a single sentence is about the ocean, which could easily have been included in section 3.1.

It could have, but we chose not to. In any case, Section A3 contains Figure A1, which provides more information about the ocean flux, but is not essential to the main text which is focused on the land sink.

Line 139: "This is designed ... spring and summer" What is the design? Is equation 1 only applied to the northern extra-tropics? Per model grid box? What is the spatiotemporal discretization of $\alpha$? If it is only applied to the northern extra-tropics than what justifies the assumption that the residual land sink in CASA is fine elsewhere? Further details are needed here.

This correction is applied everywhere and $\alpha$ is a constant. We have added some additional text here to make this clear. The adjustment "focuses" itself in the Northern Extratropics (NE) because outside of the Spring and Summer there, the term $\Delta^{+}T_{m}$ will be very close to zero. That CASA should and can be adjusted in the NE in this way to better agree with inversions is the subject of this paper.

Line 148: "about the construction and evaluation of the empirical sink, see Sections 3.2 A3" Fine to put details in A3 (even though I only found information about the ocean and biomass burning there), but evaluation section 3.2 should not deal with the construction

of the empirical land sink.

The purpose of the evaluation section is to evaluate our fluxes, notably the empirical land sink. So it's unclear why it shouldn't discuss the construction, when and where it does well, etc.

Line 152: ", yet the sink due to the corn and soybean harvest ..." This suggests that the midwest is the sink accounts for the global emission of short cycle fuels.

It very well could, but this question is beyond the scope of this paper.

Line 176: "in the Niño 3.4 region" Either this region needs to be defined, or a reference should be given where this information can be found.

We've defined the region in the text now.

Line 215: "our ocean exchange fluxes produce a sink that is generally consistent with the inversion ensemble" Whether or not this result is consistent enough depends on the requirements. I would agree that the average sink is in good agreement, however, the trend is not. There is no discussion whether or not this is important, but it seems that a NRT projection or forecast would quickly divert from the uncertainty range.

The trend is discussed in the last sentence of that same paragraph. For the 15-year period considered in this paper, our global ocean flux trends do stay within the window of the inversion ensemble. For longer time periods, using a linear pCO2sw of 1.5 muatm/yr is indeed not appropriate. However, since this growth rate is used in the construction of the pCO2sw climatology of Takahashi et al. (2009), it's unclear that simply choosing an exponential growth rate would fix the problem. Since the focus of this paper is on the land sink, we chose to leave this topic for future investigation.

It's unclear to the authors why one would expect an NRT projection or forecast to diverge quickly from the uncertainty range. Recall that the years 2016 and 2017 are forecasts and they fall within that range.

Section 3.2 The empirical land sink: According to the components specified in section 2 this does not include biomass burning and biofuel. Yet in the description of this section numbers are provided for NBE. This should be made consistent.

See line 155 from the original text:

NBE = NEE (from CASA) + empirical land sink + biomass burning + biofuel

We've made this explicit in the revision.

Line 230: "the sum the sum"

Noted and corrected.

Line 267: "adjustments to HR or NPP are both ... we look for in the empirical sink." Given this conclusion from the preceding discussion, what is it that justifies the current treatment of the empirical land sink?

The preceding discussion emphasizes that CASA, which has a neutral biosphere by design, likely lacks a sink in the Northern Extratropics during the Spring and Summer. This is why the empirical sink has the temperature increase term. We then multiplied that term by HR instead of NPP, recall NEE = HR - NPP, because we expect NPP to be better constrained by the CASA methodology.

Line 294: "This suggests that ... diagnostic fluxes with a similar skill as running a formal inversion system based on MBL data." I do not agree with this for two reasons: 1) the agreement between CT2016 and NOAA MBL sites would have been much better when using its native transport model, 2) Table 1 suggests that only MBE and Ocean fluxes from CT2016 are used. If these are combined with different anthropogenic fluxes then this would add further inconsistency. It would have been fairer to use the CarbonTracker optimized concentrations in this comparison. In particular, because the empirical land sink didn't suffer from the same transport inconsistency.

See line 281 of the original manuscript. Our run uses CT2016 fluxes for all components. We have stated this explicitly in the revision. We have also stated explicitly that the use of a different transport model likely disadvantages CT2016 in the comparison, but that the goal of a flux inversion is not to find surface fluxes that are appropriate for only one model. Although we cannot yet cite it here because it is a work in progress, the OCO-2 Model Intercomparison Project has confirmed that what we see in this initial evaluation holds in a much broader context: LoFI is comparable in skill with modern in situ inversions when evaluated against independent in situ data and TCCON retrievals. Those flux inversions were all run with their native transport models, etc.

3.4 Growth rate forecast: The authors indicate themselves that they could have extended the retrospective mode until 2018. It is not clear why this has not been done. It would have significantly strengthened the evaluation of the skill of the NRT mode (I mean by doing both modes for the 2016 – 2018 time window).

The authors feel that the evaluation is sufficient to support the scientific claims made in the paper.

Line 362: "When necessary, fluxes are downscaled to a higher resolution" It is not clear if this is done, or whether it is only a general possibility. It is also unclear which fluxes would require this step. If it is not used in the current setup then I recommend deleting this part.

It is used to do temporal downscaling from monthly to daily for NEE and spatial downscaling for pCO2sw. We agree that this was unclear and have adjusted the text accordingly.

---

## Author Comment (AC2) · 17 Nov 2020

Reviewer comments in red, responses in black.

Review: Calibrating satellite-driven carbon fluxes for retrospective and near real-time assimilation systems, by Weir et al.

Summary: The authors describe a $CO_2$ surface flux product that operates in retrospective and forecast modes, and can be provided with short latency. The product is shown to have comparable skill to full flux inversions that take much longer to calculate. Flux components such as biofuel, biomass burning, fossil fuel, and ocean flux

are taken from near real-time or published datasets. Terrestrial carbon flux is obtained from CASA-GFED; the innovation, or "hook" here is the imposition of an empirical land sink that concentrates the terrestrial sink in the northern extratropics during spring and summer, which is consistent with emerging inversion results (as opposed to placing the sink in the tropics or southern extratropics). This product, LoFI, is a suitable "prior" for inversion studies, as it has total surface flux (and spatiotemporal distribution of flux) comparable to optimized fluxes from recent inversion projects.

Review: This is a good paper. It is concise and well-written, and the product it describes has value to the scientific community. I've had the opportunity to see LOFI results presented in meetings and workshops, and I recognize this value. My formal recommendation is to accept the manuscript for publication, with minor revisions.

Initially I was a little concerned with the empirical sink, as it seemingly violates some accepted aspects of biophysics. After some thought, however, I realize that the authors are less concerned with maintaining fidelity to established physical relationships than they are with maintaining fidelity with flux inversion results, which do not take the physics into account. Inversions just say "here is what we think the flux map looks like".

Take heterotrophic respiration. There is a rich body of literature that describes how respiration increases with increasing temperature, and this is the basis for the so-called "Q10" relationships present in just about every model that simulates surface CO2 flux (CASA as well, I believe). In LoFI, they reduce respiration as temperature increases. As stated previously, I think this is tolerable because making this assumption produces the flux map that they want. However, I think the authors need to acknowledge that this assumption violates accepted biophysical theory.

The authors agree and have tried to better emphasize this in the revised text.

I'm also a little concerned about the strong MAM uptake in the Midwest crop region, shown in the third row of Figure 3. Yin et al. (2020), in a paper describing carbon uptake delay induced by floods in 2019, show (their Figure 2) show that in most years

crops aren't even planted until April or May. It's hard to believe that these regions would show a significant sink immediately after plant date.

This is a good find and something we're working on improving. Newer versions of LoFI actually mask out croplands when applying the empirical sink because of this exact problem. The version of CASA-GFED 3 that we start with maintains a robust net crop sink because it, by design, includes a corn and soybean harvest. Getting the empirical land sink to work with harvests and lateral fluxes will be a major undertaking, but it's underway. We'll note this in the revision.

Again, I'm ok with this as long as the authors acknowledge that they are trying to reproduce the maps suggested by inversions, and not doing so with a strong regard for biophysical processes. I'd like them to find a way to say "Hey, we don't care about the biophysics. The inversions tell us this is the pattern we want to have, and this is how we get it." I think this admission is important.

We agree and have tried to emphasize this in the revised paper. Our eventual goal would be to do something like this in a biophysically consistent way in, for example, CASA. This study was a first step in that direction.

Other than that, I don't have much to add. Good paper, nice read, valuable product. Good job, wish all my reviews were this easy.

Specific Comments:

Lines 152-153: The Midwest crop harvest is not a true sink. They don't take the harvest and bury it deep in the ground. The harvest is respired back, from feedlots and from people who eat food made from the harvest. This must be accounted for in models. What does CASA do about this?

There was some sloppy wording here that we thank the reviewer for pointing out. CASA removes carbon from its aboveground pools to account for the corn and soybean harvest (so that it does not respire this in the fall). That's it. Presumably we should have

some lateral flux to deal with exactly the issues that the reviewer raises, but that is an active area of research and something we hope to develop further in future versions. We've made some small changes to this paragraph to hopefully make the explanation clearer.

Line 177: Where does the land cover change map come from?

It's extrapolated from GCP (see the last sentence of that paragraph). We've added "global total" here to try and make it more clear that we just need a single number, not a spatial map.

Line 212: I like the umlaut in El Nino.

Just making sure people are still reading. This is fixed now.

Figure 1: It is hard to see LoFI in this plot. Is it directly under baseline? If the lines were thicker and the shading lighter, it would be easier to see. The scale can be shrunk too.

It's because it's identical to the baseline. We've made a note of this in the figure caption. We've kept the baseline in the ocean plots for consistency across plots. An alternative would've been to drop it in all plots, but we didn't find one preferable to the other and our choice made the figures easier to generate.

Line 230: typo

Fixed.

Lines 269-270: Boy, that discrepancy is really hard to see. Can you give us a number in the sentence describing it?

We were trying to point out the discrepancies between the blue line and the grey shading, which appear to us to be fairly visible. We've added that to the text to make sure it's clear.

Figure 4: A line at the equator might be helpful, to show how many stations are in the

Northern Hemisphere, and how many in the south. For those of us that don't have all
the stations memorized yet.

We've updated the figure to indicate what hemisphere the sites are in.

Figure A3: It is really hard to see GFED. I really had to work my old eyes to see the
tropical JJA difference.

We agree, but that's also our goal: we want QFED and GFED to be as close as possible. So the fact that you have to struggle to see these differences while the differences
with and without the empirical sink are so obvious helps us justify using QFED in place
of GFED.
* * *

---

## Author Response (AR1)

**Calibrating satellite-derived carbon fluxes for retrospective and near real-time assimilation systems**

Brad Weir1,2, Lesley E Ott2, George J Collatz2, Stephan R Kawa2, Benjamin Poulter2, Abhishek Chatterjee1,2, Tomohiro Oda1,2, and Steven Pawson2

1Universities Space Research Association, Columbia, MD, USA

2NASA Goddard Space Flight Center, Greenbelt, MD, USA

Correspondence: Brad Weir (brad.weir@nasa.gov)

Abstract. The ability to monitor and understand natural and anthropogenic variability in atmospheric carbon dioxide  $(CO_2)$  is a growing need of many stakeholders across the world. Systems that assimilate satellite observations, given their short latency and dense spatial coverage, into high-resolution global models are valuable, if not essential, tools for addressing this need. A notable drawback of modern assimilation systems is the long latency of many vital input datasets, e.g., inventories, in situ mea-

- 5 surements, and reprocessed remote-sensing data can trail the current date by months to years. This paper describes techniques for calibrating surface fluxes derived from satellite observations of the Earth's surface to be consistent with constraints from inventories and in situ  $CO_2$  datasets. The techniques are applicable in both short-term forecasts and retrospective simulations, thus taking advantage of the coverage and short latency of satellite data while reproducing the major features of long-term inventory and in situ records. Our approach begins with a standard collection of diagnostic fluxes which incorporate a variety
- 10 of remote-sensing driver data, viz. vegetation indices, fire radiative power, and nighttime lights. We then apply an empirical sink to calibrate the diagnostic fluxes to match given atmospheric and oceanic growth rates for each year. This step removes coherent, systematic flux errors that produce biases in  $CO_2$  which mask the signals an assimilation system hopes to capture. Depending on the simulation mode, the empirical sink uses different choices of atmospheric growth rates: estimates based on observations in retrospective mode and projections based on seasonal forecasts of sea surface temperature in forecasting mode.
- 15 The retrospective fluxes, when used in simulations with NASA's Goddard Earth Observing System (GEOS), reproduce marine boundary layer measurements with comparable skill to those using fluxes from a modern inversion system. The forecasted fluxes show promising accuracy in their application to the analysis of changes in the carbon cycle as they occur.

Copyright statement. The author's copyright for this publication is transferred to the National Aeronautics and Space Administration.

[revised manuscript text omitted]

$$\mathbf{S}_m = \alpha \cdot \Delta^+ T_m \cdot \mathbf{H} \mathbf{R}_m,$$

$$\Delta^+ T_m = \max(T_m - T_{m-1}, 0),$$
(1)

where Δ+Tm denotes the temperature increase from the previous month, and α is a constant scaling factor computed such that the total fluxes global total fluxes for the year match a specified atmospheric growth rate. In retrospective years (those preceding the current), we use growth rates derived from the NOAA MBL reference (Dlugokencky and Tans, 2016b), and in NRT years (the current and following) we use projections based on seasonal forecasts of sea surface temperature described in Section 2.1. For more information about the construction and evaluation of the empirical sink, see Sections 3.2 & A3.

[revised manuscript text omitted]

---

## Author Response (AR2)

*Reviewer comments in red, responses in black. When necessary for clarity, parts of the previous review are included as indented, italicized text.*

The authors believe we had misunderstood many of Reviewer 1's original comments. We hope that the responses below better address the issues this reviewer raised. Notably, 1) we changed from "calibrate" to "bias-correct" as the reviewer suggested, 2) the conclusions were rewritten to address the reviewer's concern about scientific and ACP-specific relevance, and 3) we've provided LoFI NRT comparisons below showing our results hold after 5 years of simulation.

Line 442: "The authors feel ... in our research"
This argument cannot be a justification for publication in ACP.

We have rewritten the conclusions section to give the reader more context for how the results fit into the carbon cycle literature, especially that in ACP, and what the major takeaways for simulations, flux inversion systems, and model-data evaluation are. We hope that the revised text and the responses herein, most importantly the last, address the reviewer's concern about the paper's suitability for publication in ACP.

Line 452: "The authors agree ... technically appropriate"
I don't see why "bias-correcting" would suggest a correction that is constant in time. I would rather think of a calibration as being relatively constant in time (for a good measurement device). A calibration is like a bias correction, but then of a measurement device against a commonly accepted standard. I don't see how it can apply to this case. To which flux standard is being calibrated?

The authors found this along with the previous justification compelling and have changed all instances of "calibrate" to "bias-correct".

Line 459: "The first part ... Line 91"

> *2 The LoFi flux collection: The structure of this section is unclear to me. I had expected three sections, one for the "retrospective mode" on for the "forecasting mode" preceded by everything that is common for these modes. I thought the latter was the baseline, which confusingly enough is not exactly what it turns out to be (see my next point).*
>
> *The first part of the section describes the components of the flux collection, and the second describes what we have to do differently in NRT. The baseline is simply the LoFI flux collection without the empirical land sink, as is noted in the paper on Line 91.*

I was not asking for an explanation, since I have read the paper. The comment was about the clarity of the structure in the eyes of an independent reader. From the answer I conclude that the authors chose to ignore my constructive attempt to improve the manuscript.

The authors agree that the section/subsection structure of Section 2 was somewhat confusing. We have tried to address this by sub-sectioning it as 2.1 "Individual flux components", 2.2 "Anthropogenic short-cycle burning and lateral fluxes", and 2.3 "Modifications needed for

forecasting mode". We hope that this division makes it clearer to the reader exactly what each part discusses.

Line 466: "You do understand that correctly ... of this paper"

> *Line 91: If I understand well the baseline still requires the NOAA MBL $CO_2$ measurements for the ocean flux, which would make it a "retrospective" type analysis. Some explanation is needed of the purpose of the "baseline" other than the notion that it doesn't include the empirical land sink. Initially I was assuming that it would be independent of the NOAA MBL $CO_2$ measurements, which apparently is not the case.*

> *You do understand that correctly. The main purpose of the paper was to evaluate the empirical land sink, so we kept all other products the same in the baseline package. While it would be interesting to test against an alternative baseline that does not use NOAA MBL $CO_2$ measurements in any way, testing the importance of the MBL $CO_2$ measurements, especially for constructing ocean fluxes, was not a primary objective of this paper.*

The authors do not get the point that I'm looking for a clearer description here, since I got confused with the current formulations. It needs improvement so that other readers do not need to struggle like me to understand what was done.

We have attempted to clarify this point by explicitly stating that the only difference between the baseline and LoFI fluxes is the empirical land sink.

Line 500: "The purpose ... does well, etc"
The point here was that I was missing an explanation of how the land sink was optimized. This sentence suggested that it was going to be explained in a results section, which is not the right place to explain a method.

This issue has been addressed in the revised text. Now the empirical sink component text in Section 2.1 lays out the exact equations used to compute the sink. We've also edited the text some to help avoid the possible confusion that the reviewer pointed out of the reader thinking the method would be explained later.

Line 495: "It very well could ..."

> *Line 152: ", yet the sink due to the corn and soybean harvest ..." This suggests that the midwest is the sink accounts for the global emission of short cycle fuels.*

> *It very well could, but this question is beyond the scope of this paper.*

No, clearly not. There are many other places in the world for which the same holds, so there is no reason to single out the mid-west here.

The authors misunderstood again here the question the reviewer was asking. We have now explicitly stated that our version of CASA-GFED represents the Midwestern US corn and soybean harvest but includes no other harvests. The authors agree that a global representation would be better, and this is the focus of ongoing work at NASA GSFC.

Line 512: "See line 155 ..."
The problem is still there. Section 3.2. is called "the empirical land sink", whereas it discusses NBE throughout, which is not the empirical land sink.

The authors again misunderstood the point of the reviewer's comment. We have changed the section heading to "Net Biospheric Exchange". We agree with the reviewer that the evaluation is of the NBE of LoFI, not specifically the empirical sink that we imposed.

Line 539: "The authors feel ... in the paper"
The review is suspicious that this was not shown whereas it could easily have been, because it would raise questions.

An evaluation of LoFI in NRT mode for 2015–2018 against in situ and TCCON data compared to that of modern satellite and in situ inversion systems is available at

https://www.esrl.noaa.gov/gmd/ccgg/OCO2_v9mip/
https://gmao.gsfc.nasa.gov/gmaoftp/sourish/OCO2/MIP/v2/tccon_comparison/plots/summary/

The performance of LoFI in NRT mode relative to the inversions is perhaps better in 2017 and 2018 than it was in 2015 and 2016. While inversion results are not yet available for 2019, LoFI is (a major point of this manuscript). Comparisons for 2019 LoFI NRT against in situ data show comparable performance to previous years. LoFI NRT performance is thus comparable to inversions (even at non-MBL sites) over the 5 years 2015–2019. Furthermore, the submissions in the above intercomparison are the subject of several submitted an in-preparation papers which show, among other things, that LoFI NRT runs have the best representation of the contrast of $CO_2$ across passing weather fronts as observed in NASA's ACT-America suborbital campaign. While these results are compelling, they are not included in the manuscript because they are the subject of an intercomparison-wide paper in preparation analogous to Crowell et al. (2019; doi:10.5194/acp-19-9797-2019) for a previous OCO-2 retrieval version.

One point that is important to note is that these runs are only forecasts of carbon fluxes. All runs use reanalysis meteorology from MERRA-2 which is available roughly 1 month after the current date. If we were to have forecasted the meteorology in addition to carbon fluxes, we would expect the skill to degrade within a few months and last about two years or less as demonstrated by Ilyina et al. (2020; doi:10.1029/2020GL090695). This point has also been included in Section 3.4.